# Polygenic risk score for obesity and the quality, quantity, and timing of workplace food purchases: A secondary analysis from the ChooseWell 365 randomized trial

**Hassan S. Dashti**[1,2,3], **Marie-France Hivert**[4,5], **Douglas E. Levy**[6], **Jessica L. McCurley**[7], **Richa Saxena**[1,2,3], **Anne N. Thorndike**[7] *

1 Center for Genomic Medicine, Massachusetts General Hospital and Harvard Medical School, Boston, Massachusetts, United States of America, 2 Broad Institute, Cambridge, Massachusetts, United States of America, 3 Department of Anesthesia, Critical Care and Pain Medicine, Massachusetts General Hospital and Harvard Medical School, Boston, Massachusetts, United States of America, 4 Department of Population Medicine, Harvard Medical School, Harvard Pilgrim Health Care Institute, Boston, Massachusetts, United States of America, 5 Diabetes Unit, Massachusetts General Hospital, Boston, Massachusetts, United States of America, 6 Mongan Institute Health Policy Research Center, Massachusetts General Hospital, Harvard Medical School, Boston, Massachusetts, United States of America, 7 Division of General Internal Medicine, Department of Medicine, Massachusetts General Hospital and Harvard Medical School, Boston, Massachusetts, United States of America

* athorndike@mgh.harvard.edu

**Data Availability Statement:** Data cannot be shared publicly at this time because our data contains potentially identifying participant

## Abstract

### Background

The influence of genetic risk for obesity on food choice behaviors is unknown and may be in the causal pathway between genetic risk and weight gain. The aim of this study was to examine associations between genetic risk for obesity and food choice behaviors using objectively assessed workplace food purchases.

### Methods and findings

This study is a secondary analysis of baseline data collected prior to the start of the "ChooseWell 365" health-promotion intervention randomized control trial. Participants were employees of a large hospital in Boston, MA, who enrolled in the study between September 2016 and February 2018. Cafeteria sales data, collected retrospectively for 3 months prior to enrollment, were used to track the quantity (number of items per 3 months) and timing (median time of day) of purchases, and participant surveys provided self-reported behaviors, including skipping meals and preparing meals at home. A previously validated Healthy Purchasing Score was calculated using the cafeteria traffic-light labeling system (i.e., green = healthy, yellow = less healthy, red = unhealthy) to estimate the healthfulness (quality) of employees' purchases (range, 0%–100% healthy). DNA was extracted and genotyped from blood samples. A body mass index (BMI) genome-wide polygenic score (BMI$_{GPS}$) was generated by summing BMI-increasing risk alleles across the genome. Additionally, 3 polygenic risk scores (PRSs) were generated with 97 BMI variants previously identified at the

information. The study population is a relatively small cohort of employees at Massachusetts General Hospital (a named institution in the manuscript) enrolled in the ChooseWell 365 RCT during a recent time period. Sharing information on individuals' age, sex, dates, timing of cafeteria purchases, and BMI will compromise participant privacy. Data are available from the Partners Human Research Office/Institutional Review Board at Partners HealthCare (contact located at https://www.partners.org/Medical-Research/Support-Offices/Human-Research-Committee-IRB/Default.aspx) for researchers who meet the criteria for access to confidential data.

**Funding:** ANT is supported by NHLBI R01HL125486 and NIDDK R01DK114735. ANT, RS, and HSD are supported by Massachusetts General Hospital's Center for Genomic Medicine Catalysis Award. HSD and RS are supported by NIDDK R01DK107859. RS is also supported by NIDDK R01DK102696 and MGH Research Scholar Fund. The project was supported by NIH 1UL1TR001102. The funders had no role in study design, data collection and analysis, decision to publish, or preparation of the manuscript.

**Competing interests:** The authors have declared that no competing interests exist.

**Abbreviations:** BMI, body mass index; CNS, central nervous system; CI, confidence interval; GPS, genome-wide polygenic score; GWAS, genome-wide association study; HGDP, Human Genome Diversity Project; HRC, Haplotype Reference Consortium; MGH, Massachusetts General Hospital; OR, odds ratio; PRS, polygenic risk score; SNP, single nucleotide polymorphism; STROBE, Strengthening the Reporting of Observational Studies in Epidemiology.

genome-wide significance level ($P < 5 \times 10^{-8}$): (1) $BMI_{97}$ (97 loci), (2) $BMI_{CNS}$ (54 loci near genes related to central nervous system [CNS]), and (3) $BMI_{non\text{-}CNS}$ (43 loci not related to CNS). Multivariable linear and logistic regression tested associations of genetic risk score quartiles with workplace purchases, adjusted for age, sex, seasonality, and population structure. Associations were considered significant at $P < 0.05$. In 397 participants, mean age was 44.9 years, and 80.9% were female. Higher genetic risk scores were associated with higher BMI. The highest quartile of $BMI_{GPS}$ was associated with lower Healthy Purchasing Score (−4.8 percentage points [95% CI −8.6 to −1.0]; $P = 0.02$), higher quantity of food purchases (14.4 more items [95% CI −0.1 to 29.0]; $P = 0.03$), later time of breakfast purchases (15.0 minutes later [95% CI 1.5–28.5]; $P = 0.03$), and lower likelihood of preparing dinner at home (Q4 odds ratio [OR] = 0.3 [95% CI 0.1–0.9]; $P = 0.03$) relative to the lowest $BMI_{GPS}$ quartile. Compared with the lowest quartile, the highest $BMI_{CNS}$ quartile was associated with fewer items purchased ($P = 0.04$), and the highest $BMI_{non\text{-}CNS}$ quartile was associated with purchasing breakfast at a later time ($P = 0.01$), skipping breakfast ($P = 0.03$), and not preparing breakfast ($P = 0.04$) or lunch ($P = 0.01$) at home. A limitation of this study is our data come from a relatively small sample of healthy working adults of European ancestry who volunteered to enroll in a health-promotion study, which may limit generalizability.

## Conclusions

In this study, genetic risk for obesity was associated with the quality, quantity, and timing of objectively measured workplace food purchases. These findings suggest that genetic risk for obesity may influence eating behaviors that contribute to weight and could be targeted in personalized workplace wellness programs in the future.

## Trial registration

Clinicaltrials.gov NCT02660086.

## Author summary

### Why was this study done?

- Genetics play a role in the development of obesity, yet the influence of genetic risk for obesity on food choice behaviors is not well understood.

- Workplace cafeteria purchasing data provided an opportunity for objective, real-time assessment of employees' food choices.

### What did the researchers do and find?

- We conducted a secondary analysis of baseline data of cafeteria food purchases prior to the start of a workplace intervention in a health-promotion randomized control trial using data from 397 employees who accepted to provide DNA for genetic analyses.

- We observed that the highest quartile of a genome-wide polygenic score for body mass index (highest genetic risk for obesity) was associated with lower dietary quality of all

purchases, higher quantity of food purchases, later time of breakfast purchases, and lower likelihood of preparing dinner at home relative to the lowest quartile.

### What do these findings mean?

- Genetic risk for obesity was associated with the quality, quantity, and timing of objectively measured workplace food purchases, suggesting that genetic risk for obesity may influence eating behaviors that contribute to weight.

## Introduction

Genetics play a role in the development of obesity and cardiometabolic disease [1], yet the influence of genetic risk for obesity on food choice behaviors is not well understood. The heritability of body mass index (BMI) is estimated to range from 47% to 90% [2]. BMI-associated genetic variants, either independently (i.e., *FTO*, *MC4R*) or in aggregate (i.e., in the form of a polygenic risk score [PRS]), have also been linked with self-reported eating-related traits and behaviors, including increased appetite, reduced satiety, uncontrolled eating, and emotional eating [3–6]. Self-reported behaviors, however, are prone to misreporting or social desirability biases [7]. Thus, it remains unclear whether genetic risk for obesity is associated with quality, quantity, or timing of food choices, each of which may mediate the relationship between the risk alleles and obesity.

A genome-wide association study (GWAS) in over 300,000 adult participants identified 97 common independent genetic variants that are associated with BMI [8]. More than half of these 97 variants are enriched for expression in regions of the central nervous system (CNS), including hypothalamus circuits that regulate appetite, whereas the remaining span other tissues with unlikely CNS functions [8,9]. Recently, it was demonstrated that a genome-wide polygenic score (GPS) comprising all 2.1 million common variants across the genome accounted for up to 20% of the variation in BMI [8], and this may provide a more robust score to detect obesity-related associations with dietary behaviors.

Previous large cohort studies have demonstrated that consumption of sugar-sweetened beverages accentuated genetic risk for obesity, whereas consumption of a healthier diet attenuated genetic risk for obesity [10,11]. These studies concluded that genetic risk for obesity is modified by dietary environmental exposures. However, it remains unknown if genetic risk for obesity directly influences food choice behaviors and dietary intake. Obesity-implicated genetic variants may influence eating behaviors and obesity through a range of biological mechanisms that determine taste preferences, satiety, and cognitive and physiological responses to food and food cues [11–15]. Therefore, genetic predisposition to food choice behaviors may be in the causal pathway between genetic risk and the development of obesity, and genetic variants with CNS and non-CNS functions may contribute differentially to food choice behaviors.

Workplace cafeteria purchasing data provided an opportunity for objective, real-time assessment of employees' food choices. Prior research demonstrated that the healthfulness of workplace food purchases was associated with employees' overall dietary quality and health [16]. We hypothesized that higher obesity genetic risk was associated with food purchasing patterns and self-reported dietary behaviors that contribute to weight gain and obesity.

Therefore, the aim of the current study was examine associations between obesity genetic risk and food choice behaviors using objectively assessed workplace food purchases (quality, quantity, and timing) collected at baseline from a cohort of employees who enrolled in a workplace health-promotion trial [17]. We also assessed whether higher CNS-related obesity genetic risk may be more likely to be associated with food behavioral choices than the non-CNS component.

## Methods

This study is reported as per the Strengthening the Reporting of Observational Studies in Epidemiology (STROBE) guidelines (S1 STROBE Checklist).

### Setting and participants: "ChooseWell 365" cohort

This study is a secondary analysis of baseline data collected from participants in the "Choose-Well 365" cohort, a workplace health-promotion study at Massachusetts General Hospital (MGH), prior to the start of the health-promotion intervention randomized control trial [17]. MGH is a 999-bed teaching hospital in Boston, MA, with over 27,000 employees, who are 70% female and have a mean age of 41 years. A total of 602 MGH employees (female = 79.4%; mean [standard deviation]: age = 43.6 [12.2] years; baseline BMI = 28.3 [6.5] kg/m$^2$) enrolled between September 2016 and February 2018 in the "ChooseWell 365" randomized controlled trial (Clinicaltrials.gov: NCT02660086) testing a workplace intervention to promote healthy food choices and prevent weight gain [17]. Employees were eligible for the trial if they were between 20 and 75 years of age and used their employee badge to purchase cafeteria items at least 4 times per week for at least 6 weeks during a 12-week period prior to recruitment. Additional recruitment criteria have been previously described [17]. Analyses were prospectively planned to test the associations between BMI genetic scores and food choices as outlined in the analysis plan (S1 Text). Participants provided written informed consent upon enrollment, and the study protocol was approved by Partners HealthCare Institutional Review Board (#2015P000135).

The data for the current study are restricted to participants who consented to providing genetic data and who were of European ancestry (restricted to avoid population stratification issues and to be consistent with the GWAS that were discovered in populations of European ancestry). Analyses were conducted using survey and health data collected at a baseline visit prior to the initiation of the parent trial intervention, and cafeteria data were collected retrospectively from the 3 months prior to enrollment. We used 3 months to represent typical and habitual purchases and avoid irregularities due to short-term vacations or work schedule changes. Of the 602 participants, 499 consented to have genotyping, and 397 were of European ancestry and included in the current analysis (S1 Fig).

### Workplace purchases, anthropometry, and dietary variables

During the study period, the hospital campus had 6 on-site food service locations, including 3 full-service cafeterias and 3 smaller cafes (hereafter, all referred to as "cafeterias"). The cafeterias were typically open 5 or 7 days per week and offered breakfast and lunch options in the mornings through afternoons and limited snacks/side and dinner options in the evenings and overnight. More than 1,200 different food items are available over the course of a day, including meals and entrees (e.g., hot prepared meals, prepared sandwiches and salads, and pizza), a large salad bar, snacks, and desserts, as well as hot and cold beverages. All employees in the study paid for cafeteria items by payroll deduction using their employee identification badge, and purchases were tracked using cafeteria sales data [16].

All hospital cafeterias labeled food and beverages with traffic-light labels, as has been described in detail elsewhere [17,18], and this labeling system had been utilized at the hospital since 2010. Briefly, the traffic-light labeling system was designed by hospital nutrition staff and based on the USDA Dietary Guidelines [19,20]. Every item was labeled as red, yellow, or green based on an algorithm that factored in calories, saturated fat content, and nutrient density. A green rating connoted the highest level of healthfulness and a red rating indicated the lowest level (e.g., least healthy). Options were distributed roughly evenly between those labeled green (34%), yellow (37%), or red (29%). The average costs of red, yellow, and green items were comparable for beverages, entrees, and snacks/side items, and items across a range of prices were available in each color category [16].

Participants' baseline purchases were extracted from the hospital's cafeteria sales data for the 3 months prior to their enrollment in the randomized trial. Purchasing data included item type, time and date of purchase, and the traffic-light label color (i.e., red, yellow, green). The quality of workplace food purchases was measured with a Healthy Purchasing Score that reflected the overall healthfulness of an employee's 3-month baseline purchases [16]. The Healthy Purchasing Score was created by weighting purchases of red items to be 0, yellow items to be 0.5, and green items to be 1. This score has been previously validated as a proxy for overall dietary healthfulness using 24-hour dietary recalls [16]. For interpretation purposes in this study, the Healthy Purchasing Score was converted to percentage by multiplying the score by 100 (range, 0%–100% healthy). The quantity of workplace food purchases was measured by the total number of items purchased over 3 months, as well as the number of food and beverage items, separately. Time of day of workplace food purchases was measured by using the time stamp for the purchase data during the 3-month baseline period. The median timing of breakfast purchases (first food purchased between 6 AM and 10 AM) and median time of lunch purchase (first food purchased between 11 AM and 2 PM) were estimated for each participant.

Participants' weight and height were measured by clinical research nursing staff at the baseline visit, and BMI was calculated as weight/height$^2$ (kg/m$^2$). Participants also completed an online survey that provided self-reported age, sex, meal-skipping habits ("Over the PAST WEEK, on how many days did you SKIP BREAKFAST/ LUNCH/ DINNER for any reason?"; Never, 1–2 days, 3–4 days, 5–6 days, and Every day), and home-prepared meal habits ("Over the PAST WEEK, on how many days did you eat a BREAKFAST/ LUNCH/ DINNER that was prepared at home [including meals that you bring to work]?"; Never, 1–2 days, 3–4 days, 5–6 days, and Every day).

## Genetic data genotyping, imputation, and quality control

DNA was extracted from blood samples collected from 499 participants and genotyped using the Infinium Global Screening (GSA) Array-24 v2.0. Imputation was performed using the Michigan Imputation server with the Haplotype Reference Consortium (HRC, Version r1.1 2016) reference panel for imputation [21]. This HRC panel consists of 64,940 haplotypes of predominantly European ancestry. Haplotype phasing was performed using Eagle v2.3 [22]. Low-quality genetic markers in Hardy-Weinberg disequilibrium ($P < 10^{-6}$), low minor allele frequency (<0.01), and low call rate (<98%) were excluded (200,067 genetic markers excluded). Furthermore, samples were tested for low-quality genetic samples with low sample call rate (<95%) or high heterozygosity rate (>median + 3*IQR), but none were excluded.

Participant ancestry was determined using TRACE [23] and the Human Genome Diversity Project (HGDP) [24] as a reference panel. Principal component analysis outliers were determined by using a principal component analysis projection of the study samples onto the

HGDP reference samples and were subsequently excluded from analysis ($n = 100$ excluded). To correct for population stratification, we computed principal components of ancestry using TRACE [23] in the subset with genetically European ancestry. Furthermore, sample relatedness was determined using PLINK [25], and subsequently, 1 sample from each detected related pair (pi-hat $> 0.25$) was excluded.

## Generation of PRSs

A total of 3 PRSs were generated for each participant from 97 previously identified single nucleotide polymorphisms (SNPs) at the genome-wide significance levels ($P < 5 \times 10^{-8}$). The BMI$_{97}$ PRS comprised all 97 previously identified BMI variants [8]. Based on the biological functions of genes in or near the 97 previously identified BMI loci, such as neuronal development process, neurotransmission, hypothalamic expression and regulatory function, and neuronal expression, 54 variants have been previously classified as CNS-related, and 43 variants have been previously classified as non-CNS-related [8]. Accordingly, the BMI$_{CNS}$ PRS and the BMI$_{non-CNS}$ PRS comprised the 54 and 43 non-overlapping BMI variants, respectively [26]. All SNPs had a minor allele frequency $>1\%$ and an imputation quality (minimac $r_{sq}$) $\geq 0.50$. We derived the PRSs for each individual participant by summing the number of risk alleles that were each weighted by the allelic effect sizes (β-coefficients) published in the original GWAS meta-analysis with up to 339,224 individuals from 125 studies [8]. Scaling of the individual PRSs was performed to allow interpretation of the effects as a per-1 risk allele increase in the PRS for each trait (division by twice the sum of the β-coefficients and multiplication by twice the square of the SNP count representing the maximum number of risk alleles).

## Generation of GPS

We generated a BMI GPS for each individual by summing BMI-increasing risk alleles across the genome, each weighted by the beta estimate for that allele from the BMI GWAS meta-analyses [8]. Only SNPs with a minor allele frequency $>1\%$ and an imputation quality (minimac $r_{sq}$) $\geq 0.50$ were considered in the GPS. Thus, we included 1,988,363 SNPs after excluding X chromosome variants and, at each site, clumped SNPs based on association $P$ value (the variant with the smallest $P$ value within a 250-kb range was retained and all those in linkage disequilibrium, $r^2 > 0.1$, were removed). Linkage disequilibrium clumping and GPS generation were conducted using PRSice [27], and the best-fit genome-wide BMI GPS based on this cohort's inverse normalized BMI encompassed 64,952 SNPs at $P$ value threshold of 0.19. In sensitivity analyses, we also generated BMI GPS based on other $P$ value thresholds (1.00 [SNPs $n = 126,161$], 0.50 [SNPs $n = 98,995$], and 0.25 [SNPs $n = 73,412$]) and re-ran analyses.

## Statistical analysis

Non-normally distributed outcome variables (BMI and purchase data) were inverse normalized prior to analysis. Breakfast-, lunch-, and dinner-skipping variables from surveys were dichotomized to daily eaters (never skip) and skippers (skip 1 or more meal per week). Breakfast, lunch, and dinner "prepared at home" variables were dichotomized to fewer than 3 days of home meals per week or 3 or more days of home meals per week. Genetic score quartiles (e.g., Q1 = lowest genetic risk, Q4 = highest genetic risk) were generated based on the population distribution, with the highest quartile representing the more adverse phenotype (higher BMI). As enrollment was year-round, sine and cosine functions of the date of enrollment were used to adjust for seasonality in the participants' 3-month purchasing periods [28]. The GPS was standardized to have a mean of 0 and a standard deviation of 1. Multivariable linear regression was used to test the association of each continuous, scaled PRS or GPS quartiles

with BMI adjusted for age, sex, and 5 principal components of ancestry (identified by TRACE). In addition, multivariable linear or logistic regression was used to test genetic score associations with workplace purchases and survey-derived meal habits adjusted for age, sex, seasonality, and 5 principal components of ancestry. Differences across quartiles were evaluated for significance using a test for trend. We tested for statistical interaction between CNS-related and non-CNS-related results from stratified analyses ($P_{int}$). Associations with purchases and meal habits were also repeated using continuous measures of the genetic scores. We present unadjusted ($P$) and false discovery rate–corrected $P$ values ($P_{adj}$) to account for multiple testing. In sensitivity analyses, we further adjusted for job type (administrative/service, craft/technicians, management/professionals, MDs/PhDs), education level (high school/some college, college degree, graduate degree), current smoking status, and physical activity level (measured with the International Physical Activity Questionnaire [29] at the baseline visit). All genetic analyses were conducted in R version 3.6.2 (2019 December 12), and associations were considered significant at $P < 0.05$.

## Results

Genetic analyses were restricted to 397 unrelated participants of European ancestry with high-quality genetic data in the "ChooseWell 365" study (Table 1, S1 Fig). The mean age of participants included in this analysis was 44.9 years, and 80.9% were female. The medians (ranges)

Table 1. General characteristics of "ChooseWell 365" study participants of European ancestry ($n$ = 397).

| Characteristics | Mean (SD) or Percentage |
| --- | --- |
| Age, years | 44.9 (12.8) |
| Sex, % female | 80.9 |
| Body mass index, kg/m$^2$ | 27.9 (6.1) |
| Job type, % | |
| Administrative/service | 9.1 |
| Craft/technicians | 9.3 |
| Management/professionals | 71.3 |
| MDs/PhDs | 10.3 |
| Education level, % | |
| High school/some college | 8.1 |
| College degree | 43.1 |
| Graduate degree | 48.4 |
| Smoking status (% current) | 2.5 |
| Physical activity*, % | |
| Low | 2.3 |
| Moderate | 28.0 |
| High | 69.7 |
| Skips breakfast >1 day/week, % | 41.3 |
| Skips lunch >1 day/week, % | 34.5 |
| Skips dinner >1 day/week, % | 20.9 |
| Breakfast prepared at home ≥3 days/week, % | 28.7 |
| Lunch prepared at home ≥3 days/week, % | 9.6 |
| Dinner prepared at home ≥3 days/week, % | 54.9 |

*Physical activity measured with the International Physical Activity Questionnaire at the baseline visit.

Abbreviations: SD, standard deviation

for the number of BMI-increasing alleles observed were 90 (70–105) for $BMI_{97}$, 57 (42–70) for $BMI_{CNS}$, and 33 (22–44) for $BMI_{non-CNS}$ (S2 Fig).

The GPS ($BMI_{GPS}$) accounted for 14.8% of variance in BMI. The 97 loci ($BMI_{97}$) accounted for 2.2% of variance in BMI, consistent with earlier reports [8], and the $BMI_{CNS}$ and $BMI_{non-CNS}$ accounted for 1.2% and 0.5% of the BMI variance, respectively. Of the 97 BMI loci, 56 signals showed a direction of association consistent with the discovery GWAS (binomial $P = 0.03$) (S1 Table). Generally, higher quartiles of the genetic scores were consistently associated with higher BMI (Fig 1). The highest quartile (Q4) of the $BMI_{GPS}$ was associated with a 6.4-kg/m$^2$-higher (95% confidence interval [CI] 4.8–8.0) BMI relative to the lowest quartile (Q1) ($P = 2.1 \times 10^{-15}$). By comparison, the highest quartiles of the $BMI_{97}$, $BMI_{CNS}$, and $BMI_{non-CNS}$ PRS were associated with a 2.4-kg/m$^2$-higher (95% CI 0.7–4.1; $P = 0.01$), 1.9-kg/m$^2$-higher (95% CI 0.2–3.6; $P = 0.10$), and 1.9-kg/m$^2$-higher (95% CI 0.2–3.6; $P = 0.04$) BMI relative to the lowest quartile, respectively.

Fig 2 shows associations between the $BMI_{GPS}$ and the $BMI_{97}$ genetic scores and workplace purchases (quality, quantity, and timing). The highest quartile of the $BMI_{GPS}$ was associated with a lower Healthy Purchasing Score relative to the lowest quartile of the $BMI_{GPS}$ (−4.8 percentage points [95% CI −8.6 to −1.0]; $P = 0.02$). The highest quartile of the $BMI_{GPS}$ was also associated with purchasing more food items over the 3-month period (14.4 more items [95% CI −0.1 to 29.0]; $P = 0.03$) and with purchasing breakfast later (15.0 minutes later [95% CI 1.5–28.5]; $P = 0.03$) than the lowest quartile of $BMI_{GPS}$. There were no significant associations between the $BMI_{97}$ and workplace food purchases. Fig 3 shows associations between purchases and $BMI_{CNS}$ and $BMI_{non-CNS}$ risk scores. Significant heterogeneity between CNS and non-CNS was observed for the associations for total purchases ($P_{int} = 0.02$), food purchases ($P_{int} = 0.03$), and breakfast timing ($P_{int} = 0.04$). Relative to the lowest quartile, the highest $BMI_{CNS}$ quartile was associated with fewer total (−18.5 items [95% CI −38.4 to 1.4]; $P = 0.04$) and food items (−12.6 food items [95% CI −27.2 to 2.0]; $P = 0.04$) purchased at work, and the highest $BMI_{non-CNS}$ quartile was associated with purchasing breakfast later than the lowest quartile (17.9 minutes later [95% CI 4.5–31.3]; $P = 0.01$).

Associations between the BMI genetic scores and self-reported meal skipping or meals prepared at home are demonstrated in Fig 4 and Fig 5. Higher $BMI_{GPS}$ was associated with lower odds of preparing dinner at home (Q4 odds ratio [OR] = 0.3 [95% CI 0.1–0.9]; $P = 0.03$), and no significant associations between the $BMI_{97}$ genetic scores and meal skipping or meals prepared at home were observed (Fig 4). In addition, $BMI_{non-CNS}$ was associated with higher odds of skipping breakfast (Q4 OR = 2.0 [95% CI 1.1–3.7]; $P = 0.03$) and lower odds of preparing breakfast (Q4 OR = 0.5 [0.3–1.0]; $P = 0.04$) or lunch (Q4 OR = 0.4 [0.2–0.8]; $P = 0.01$) at home, with significant heterogeneity between CNS and non-CNS associations observed for preparing lunch at home only ($P_{int} = 0.004$) (Fig 5).

Accounting for multiple testing resulted in $P_{adj}$ greater than 0.05 for all findings (S2 Table). In sensitivity analyses, results were similar when BMI genetic scores were expressed as continuous measures (S3 Table); when these models were further adjusted for job type, education level, smoking status, and physical activity level in sensitivity analyses (S4 Table); and when the $BMI_{GPS}$ was calculated using other $P$ value thresholds (S5 Table).

## Discussion

In this study, we used objective, real-time food purchasing data to investigate the association of food choice behaviors with genetic risk for obesity. Using genetic risk scores derived with variants from a previous BMI GWAS meta-analysis, we found evidence that employees' genetic risk was associated with the quality, quantity, and timing of the food they purchased at

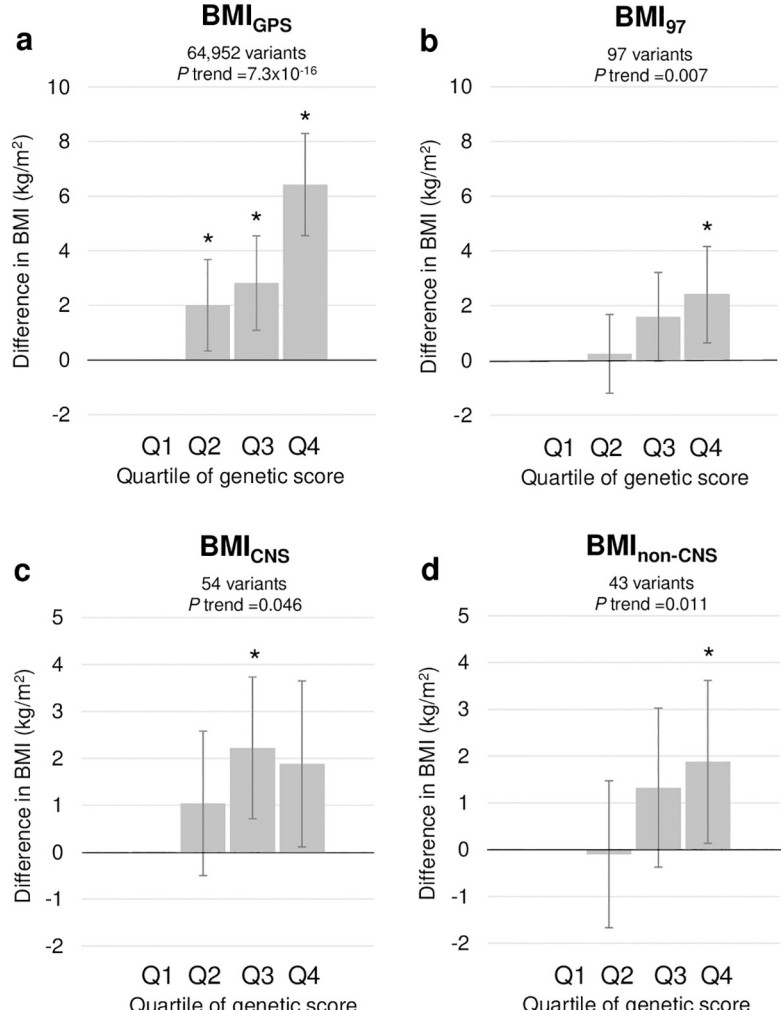

**Fig 1. Association of BMI genetic scores with participants' BMI (in kg/m²) according to quartiles of genetic scores.** Higher quartiles reflect more BMI-increasing alleles. y-Axis is difference in BMI (in kg/m²) compared with reference quartile (Q1) adjusted for age, sex, and 5 principal components of ancestry. Asterisks denote significant (i.e., $P < 0.05$) difference between quartile and Q1. The BMI$_{GPS}$ is a GPS comprising 36,172 BMI-increasing risk alleles across the entire genome. The BMI$_{97}$ PRS is restricted to 97 previously identified BMI variants at the genome-wide threshold [8]. Based on the biological functions of genes in or near the 97 previously identified BMI loci, the BMI$_{CNS}$ PRS and BMI$_{non-CNS}$ PRS comprise 54 variants previously classified as CNS-related and 43 variants previously classified as non-CNS-related, respectively [8]. For interpretation purposes, difference in BMI is derived from models in which BMI is untransformed, whereas $P$ values are derived from models in which BMI is inverse normalized. BMI, body mass index; CNS, central nervous system; GPS, genome-wide polygenic score; OR, odds ratio; PRS, polygenic risk score.

work. Higher genetic risk was associated with several workplace food choice behaviors that may contribute to weight gain and obesity, including purchasing less healthy food; purchasing larger quantities of food; purchasing meals at later times; skipping breakfast; and being less likely to prepare meals at home. Prior research has demonstrated that lifestyle modification may attenuate genetic susceptibility to obesity and cardiometabolic risk [11,30,31], and therefore our preliminary findings may have important implications for tailoring health-promotion and workplace wellness programs in the future.

Overall, BMI$_{GPS}$ had the strongest associations with BMI and food choice behaviors. Higher BMI$_{GPS}$ was associated with a lower Healthy Purchasing Score, a measure of the dietary quality

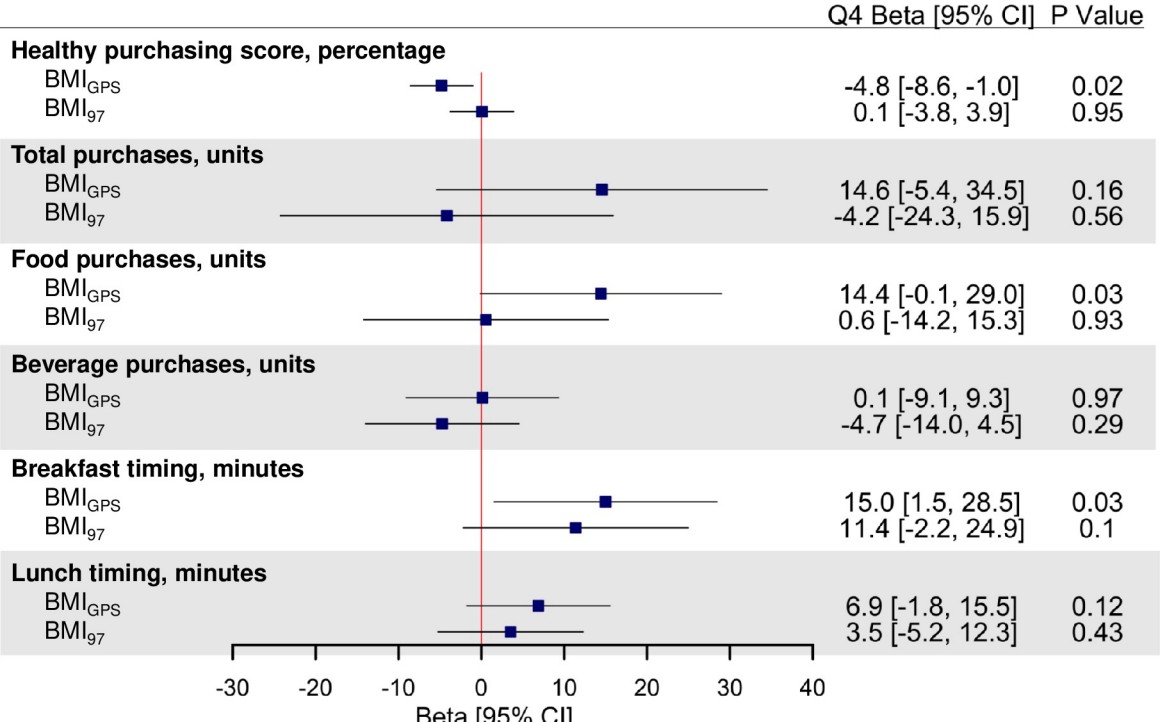

**Fig 2. BMI genetic scores associations with quality, quantity, and timing of workplace purchases.** Association results are adjusted betas reflecting difference in Healthy Purchasing Score (percentage), items purchased (units over 3-month period), or timing (in minutes) between highest (Q4) and lowest (Q1, reference) quartile of BMI genetic scores adjusted for age, sex, seasonality, and 5 principal components of ancestry. Higher purchasing score = healthier purchases (0%–100%). For interpretation purposes, adjusted betas are from models in which outcomes are untransformed, whereas $P$ values are derived from models in which outcomes are inverse normalized. $P$ values are unadjusted for multiple testing, and false discovery rate–corrected $P$ values ($P_{adj}$) are presented in S2 Table. BMI, body mass index; CI, confidence interval; GPS, genome-wide polygenic score.

of workplace food that has been correlated with overall dietary quality, as measured by 24-hour dietary recalls [16]. The reasons for associations between higher genetic risk and purchasing more unhealthy foods at work are likely multifactorial and may include individual preferences for less healthy foods (e.g., foods high in saturated fat or sugar) [15], impulsive behavior [32], and vulnerability to unhealthy cues in the food environment [33]. These associations are consistent with prior studies demonstrating that higher numbers of *FTO* obesity-related risk alleles were associated with more eating episodes per day, higher calories consumed at lunch, and stronger responses to food cues [14,34,35]. Other loci have established roles in anorexigenic and orexigenic signaling pathways (*BDNF*) [36], nutrient preference such as carbohydrate (*RARB*) [6] and fat (*ADH1B*) [37], and energy homeostasis (*MTCH2*) [38]. Our findings, however, do not implicate specific genes or mechanisms, and the precise biological role of most BMI loci remain to be elucidated [11].

Higher BMI$_{GPS}$ was also associated with purchasing more food items at work and being less likely to prepare dinner at home, a behavior previously associated with obesity in adults [39]. Lastly, the association between higher genetic risk and later breakfast purchases corroborates earlier nongenetic epidemiological findings between higher BMI and later food intake [40,41]. The greater number of significant associations observed for the BMI$_{GPS}$ relative to the BMI$_{97}$ may be due to the fact that the BMI$_{GPS}$ explained 6 times more BMI variance than the BMI$_{97}$, resulting in a more robust genetic risk score and possibly capturing more of the underlying biology for BMI. Several findings for the BMI$_{97}$ were consistent in direction with the BMI$_{GPS}$

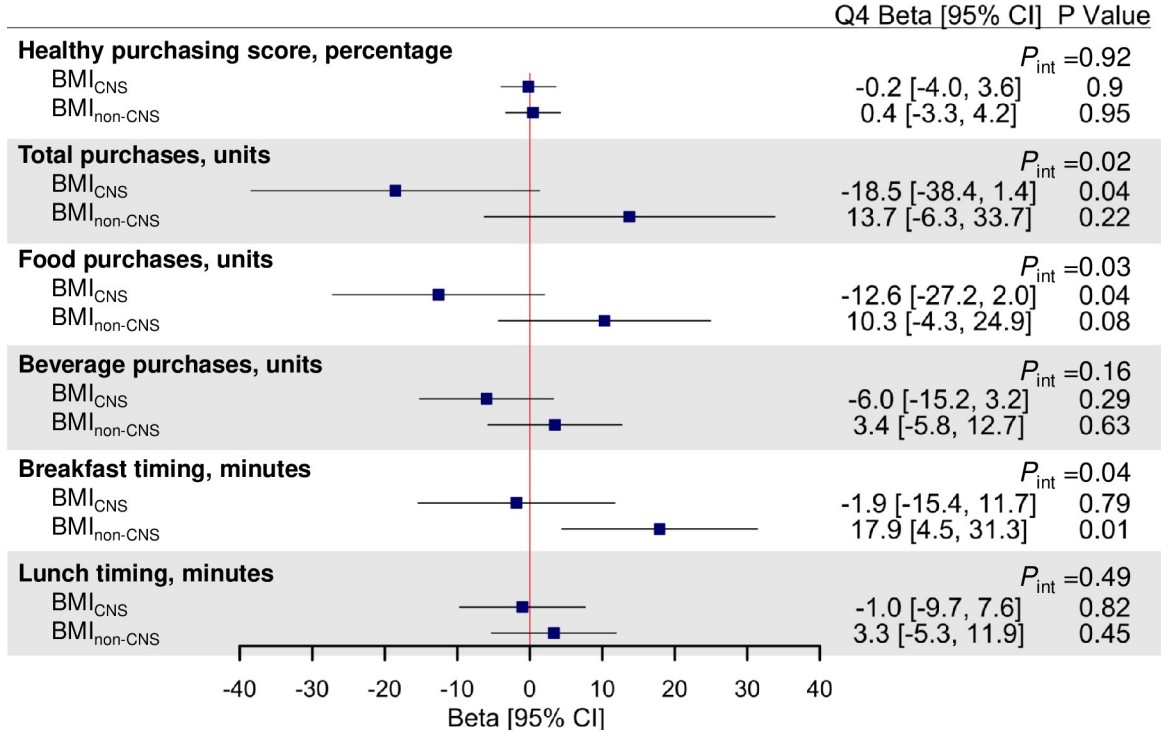

**Fig 3. BMI_CNS and BMI_non-CNS genetic scores associations with quality, quantity, and timing of workplace purchases.** Association results are adjusted betas reflecting difference in Healthy Purchasing Score (percentage), items purchased (units over 3-month period), or timing (in minutes) between highest (Q4) and lowest (Q1, reference) quartile of BMI genetic scores adjusted for age, sex, seasonality, and 5 principal components of ancestry. Higher purchasing score = healthier purchases (0%–100%). For interpretation purposes, adjusted betas are from models in which outcomes are untransformed, whereas *P* values are derived from models in which outcomes are inverse normalized. *P* values are unadjusted for multiple testing, and false discovery rate–corrected *P* values (*P*_adj) are presented in S2 Table. Based on the biological functions of genes in or near the 97 previously identified BMI loci, the BMI_CNS PRS and BMI_non-CNS PRS comprise 54 variants previously classified as CNS-related and 43 variants previously classified as non-CNS-related, respectively. BMI, body mass index; CI, confidence interval; CNS, central nervous system; GPS, genome-wide polygenic score; int, interaction; PRS, polygenic risk score.

and may require a larger sample size before significant associations are detected. Collectively, these findings suggest that genetic risk for obesity may play a role in vulnerability to food behaviors that contribute to weight gain.

To further explore the association of genetic risk for obesity and food choice, we examined differences by the 97 genetic variants for BMI that are enriched for expression in regions of the CNS compared with the remaining variants that are expressed in other tissues with non-CNS functions. Prior research provided evidence indicating possible differential effects of these subsets, with BMI_CNS having a stronger interaction with dietary quality on BMI than BMI_non-CNS [11]. In our study, we observed subset differences in the quantity of food purchased at work, with higher BMI_CNS associated with purchasing fewer food items at work and higher BMI_non-CNS trending toward purchasing more food items at work. Higher BMI_non-CNS was associated with purchasing breakfast at a later time, skipping breakfast, and not preparing breakfast or lunch at home, but BMI_CNS was not associated with any of these factors. Contrary to our initial hypothesis, our results suggest that BMI_non-CNS had a stronger association with employees' food choice behaviors that could lead to weight gain than BMI_CNS. Given that the workplace food environment in this study utilized cues to promote healthier eating (i.e., traffic-light labels), it is possible that these cues may have attenuated the unhealthy food choice behaviors of employees with higher BMI_CNS but did not interact with behaviors of employees with higher

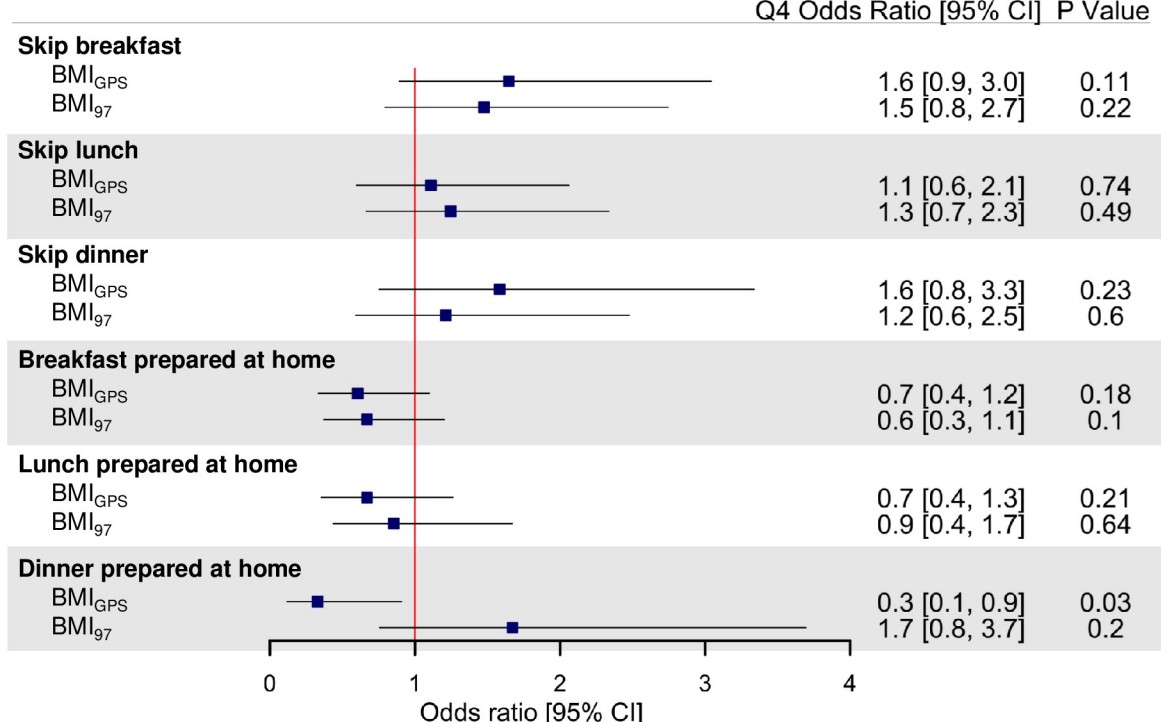

**Fig 4. BMI genetic scores and self-reported meal skipping and meals prepared at home.** Association results are adjusted odds ratio comparing highest (Q4) with lowest (Q1) quartile of BMI genetic scores adjusted for age, sex, seasonality, and 5 principal components of ancestry. Odds ratio >1 indicates more meal skipping or more meals prepared at home. *P* values are unadjusted for multiple testing, and false discovery rate–corrected *P* values ($P_{adj}$) are presented in S2 Table. BMI, body mass index; CI, confidence interval; GPS, genome-wide polygenic score.

$BMI_{non-CNS}$. Although these preliminary findings will need to be confirmed in larger samples, our results may have implications for tailoring interventions for subgroups of individuals, and insights could be leveraged to unravel and classify subtypes of BMI genetic risk, as has been conducted for other diseases such as diabetes [42].

A major strength of this study is the objective and comprehensive assessment of workplace food purchases derived from 3 months of cafeteria sales data. These measures were not prone to misreporting or social desirability biases, in contrast to many prior studies in the field. Our approach also enabled a multidimensional capture of food choice behavior, including timing. Furthermore, the relevant self-reported dietary behavior provided results that were complementary to the cafeteria purchasing findings.

There are also important limitations. Although the cafeteria systems provided objective data, food purchases may not have reflected actual food consumption that may have been influenced by work shift schedules and cafeteria hours. The $BMI_{CNS}$ and $BMI_{non-CNS}$ designations were limited to the 97 previously characterized BMI genetic variants, but additional variants may be missing from these genetic scores. We primarily present *P* values unadjusted for multiple testing because of our modest sample size and because the BMI genetic risk scores and the outcomes tested were not independent. Therefore, we acknowledge that individual "significant" findings should be interpreted with caution, since accounting for multiple comparisons led to nonsignificant *P* values, and that larger samples are necessary to verify our findings. Although the types of data we collected suggest we could infer potential causal links using mendelian randomization analyses, our sample size is too small for such an approach.

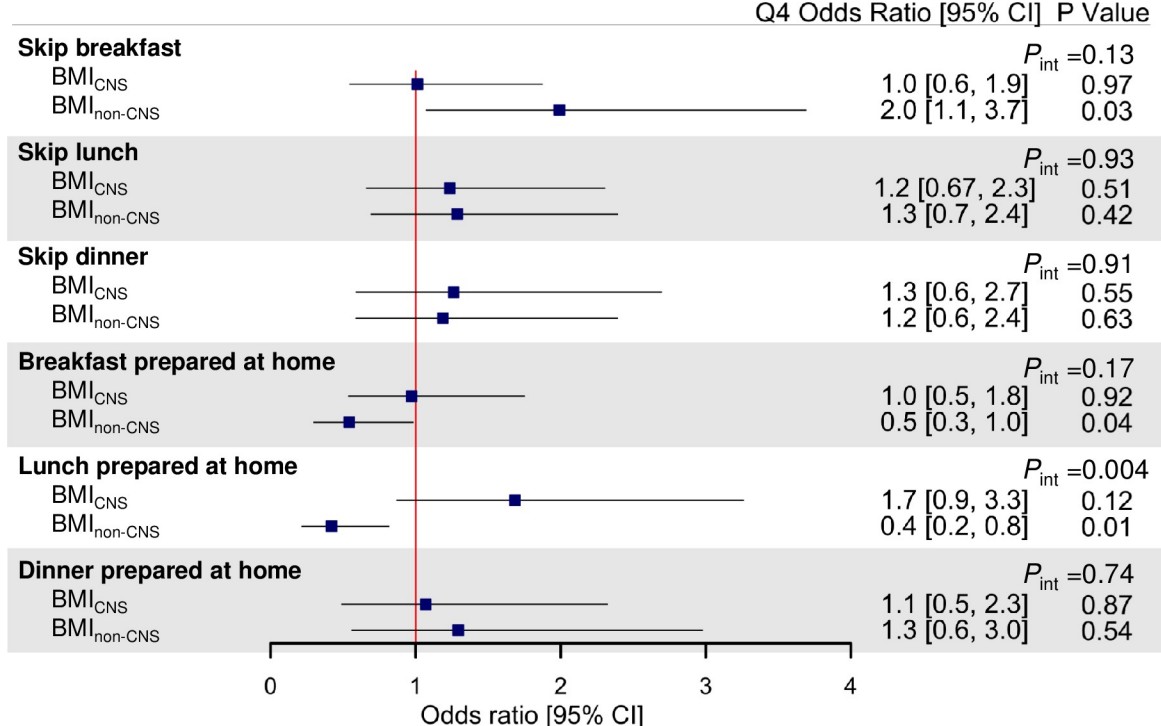

**Fig 5. BMI_CNS and BMI_non-CNS genetic scores and self-reported meal skipping and meals prepared at home.** Association results are adjusted odds ratio comparing highest (Q4) with lowest (Q1) quartile of BMI genetic scores adjusted for age, sex, seasonality, and 5 principal components of ancestry. Odds ratio >1 indicates more meal skipping or more meals prepared at home. *P* values are unadjusted for multiple testing, and false discovery rate–corrected *P* values ($P_{adj}$) are presented in S2 Table. BMI, body mass index; CI, confidence interval; CNS, central nervous system; GPS, genome-wide polygenic score; int, interaction.

Selection bias was possible as a result of inclusion and exclusion criteria for the randomized trial in addition to the criteria for this analysis, such as non-European ancestry (restricted to avoid population stratification issues). Also, despite our covariate adjustment, we recognize that a general weakness of observational studies is a risk of bias due to residual confounding. Finally, our sample consisted of a relatively small number of healthy working adults at a large urban hospital who had volunteered to enroll in a health-promotion study, which may limit generalizability of our findings to other working populations and rural or non-employed people.

In conclusion, this study identified associations between obesity genetic risk scores and food choice behaviors, suggesting that the genetic risk for obesity may play a role in vulnerability to food behaviors that are relevant for the development of obesity. Our findings demonstrated that higher genetic risk for BMI was associated with workplace food choice behaviors that may contribute to weight gain. Prior research has shown that healthy lifestyle behaviors, including dietary intake, can attenuate weight gain and cardiovascular disease in those at high genetic risk [11,30,31]. Therefore, understanding genetic predisposition to certain food choice behaviors that contribute to cardiometabolic disease could inform interventions that are tailored to changing individuals' dietary habits.

## Supporting information

**S1 STROBE Checklist. STROBE checklist.** STROBE, Strengthening the Reporting of Observational Studies in Epidemiology.
(DOCX)

**S1 Text. Prospective analysis plan.**
(DOCX)

**S1 Table. Individual 97 BMI single nucleotide polymorphism associations with BMI (in kg/m$^2$) in the "ChooseWell 365" study ($n$ = 397).** BMI, body mass index.
(DOCX)

**S2 Table. Unadjusted ($P_{unadj}$) and false discovery rate–corrected ($P_{adj}$) $P$ values for BMI genetic scores associations with workplace purchases and self-reported meal skipping and meals prepared at home.** BMI, body mass index.
(DOCX)

**S3 Table. BMI genetic scores as continuous measures associations with workplace purchases and self-reported meal skipping and meals prepared at home.** BMI, body mass index.
(DOCX)

**S4 Table. Sensitivity analyses for BMI genetic scores associations with workplace purchases and self-reported meal skipping and meals prepared at home further adjusted for job type, education level, smoking status, and physical activity level.**
(DOCX)

**S5 Table. Sensitivity analyses for BMI GPS associations with workplace purchases and self-reported meal skipping and meals prepared at home based on other $P$ value thresholds ($P$ = 0.25; 0.50; 1.00).** BMI, body mass index; GPS, genome-wide polygenic score.
(DOCX)

**S1 Fig. Flow chart of included "ChooseWell 365" study participants in present analysis.**
(DOCX)

**S2 Fig.** Count of BMI-increasing alleles for (a) BMI$_{97}$, (b) BMI$_{CNS}$, and (c) BMI$_{non-CNS}$. BMI, body mass index; CNS, central nervous system.
(DOCX)

## Author Contributions

**Conceptualization:** Hassan S. Dashti, Marie-France Hivert, Douglas E. Levy, Richa Saxena, Anne N. Thorndike.

**Data curation:** Jessica L. McCurley, Anne N. Thorndike.

**Formal analysis:** Hassan S. Dashti.

**Investigation:** Hassan S. Dashti, Anne N. Thorndike.

**Methodology:** Hassan S. Dashti, Marie-France Hivert, Douglas E. Levy, Richa Saxena, Anne N. Thorndike.

**Resources:** Anne N. Thorndike.

**Software:** Hassan S. Dashti.

**Supervision:** Anne N. Thorndike.

**Validation:** Anne N. Thorndike.

**Visualization:** Anne N. Thorndike.

**Writing – original draft:** Hassan S. Dashti, Anne N. Thorndike.

**Writing – review & editing:** Hassan S. Dashti, Marie-France Hivert, Douglas E. Levy, Jessica L. McCurley, Richa Saxena.

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
