## [Editor Report · Decision Letter 0]

6 Feb 2020

Dear Dr Dashti, 

Thank you for submitting your manuscript entitled "Association of genetic risk for obesity with the quality, quantity, and timing of workplace food purchases" for consideration by PLOS Medicine.

Your manuscript has now been evaluated by the PLOS Medicine editorial staff [as well as by an academic editor with relevant expertise] and I am writing to let you know that we would like to send your submission out for external peer review.

Kind regards,

Adya Misra, PhD,

Senior Editor

PLOS Medicine

---

## [Decision Letter · Decision Letter 1]

15 May 2020

Dear Dr. Dashti,

Thank you very much for submitting your manuscript "Association of genetic risk for obesity with the quality, quantity, and timing of workplace food purchases" (PMEDICINE-D-20-00219R1) for consideration at PLOS Medicine. 

[LINK]

In light of these reviews, I am afraid that we will not be able to accept the manuscript for publication in the journal in its current form, but we would like to consider a revised version that addresses the reviewers' and editors' comments. Obviously we cannot make any decision about publication until we have seen the revised manuscript and your response, and we plan to seek re-review by one or more of the reviewers. 

We expect to receive your revised manuscript by May 29 2020 11:59PM. Please email us (plosmedicine@plos.org) if you have any questions or concerns.

We look forward to receiving your revised manuscript. 

Sincerely,

Adya Misra, PhD

Senior Editor 

PLOS Medicine

plosmedicine.org

Please revise your title according to PLOS Medicine's style. Your title must be nondeclarative and not a question. It should begin with main concept if possible. "Effect of" should be used only if causality can be inferred, i.e., for an RCT. Please place the study design ("A randomized controlled trial," "A retrospective study," "A modelling study," etc.) in the subtitle (ie, after a colon).

Abstract

Aim and study design should be clearly stated in the background and methods section respectively

Please provide brief participant demographics and when the study took place

Please provide participant numbers

Please provide precise p-values, except when p<0.001

The last sentence of the methods and findings section must highlight a limitation of your study design or methodology

Abstract Conclusions:

* Please address the study implications without overreaching what can be concluded from the data; the phrase "In this study, we observed ..." may be useful.

* Please interpret the study based on the results presented in the abstract, emphasizing what is new without overstating your conclusions.

* Please avoid vague statements such as "these results have major implications for policy/clinical care". Mention only specific implications substantiated by the results.

* Please avoid assertions of primacy ("We report for the first time....")

Author summary

Prospective analysis plan

Did your study have a prospective protocol or analysis plan? Please state this (either way) early in the Methods section.

Introduction

Please conclude this section with a clearly outlined aim of your study

Methods

Please use the Methods heading instead of “subjects and methods”. 

Since this is a secondary analysis of an RCT, please do explicitly mention this in the abstract and methods sections 

Results

Please provide precise p values

Line 310-311- we do not permit instances of data not shown, so please provide these data or remove this sentence 

Please report your study according to the relevant guideline (http://www.equator-network.org/), perhaps STROBE/STREGA guideline, and include the completed checklist as Supporting Information. When completing the checklist, please use section and paragraph numbers, rather than page numbers. Please add the following statement, or similar, to the Methods: "This study is reported as per the Strengthening the Reporting of Observational Studies in Epidemiology (STROBE) guideline (S1 Checklist)." Please report your study according to the relevant guideline, which can be found here: http://www.equator-network.org/

Comments from the reviewers:

Reviewer #1: The authors present a very interesting study that correlates genetic risk scores for BMI with food choices.

The genetic risk scores are generated in four different ways based on an existing GWAS, one genetic risk score is based on genome-wide data (GRS), while the other three scores are based on 97 genome-wide significant SNPs, which is further subdivided into 54 SNP related to the central nervous system and 43 SNPs which are not.

The design of the study is innovative and could provide important insights into how genetics influence human behaviour. The manuscript is well written and clear. Yet, I would appreciate if the authors could address my comments on the statistical analysis.

Major comments:

1. Different results depending on the risk score used: How do the authors explain the stark difference in results between the GRS and 97 SNP risk scores as shown in Figure 2? While there are suggestive findings for GRS, the results for the 97 SNP risk score are Null.

2. Overfitting of the genome-wide PRS: How robust are the results with respect to different p-value cut-offs of the genome-wide PRS? A p-value threshold of 0.19 seems rather lenient and arbitrary. Did the authors perform any more sensitivity analysis on this p-value threshold? May the tuning of the threshold parameter in the actual dataset induce overfitting?

3. Multiple testing: Why did the study not adjust for multiple testing? In my opinion this study would require to adjust at least for the number of outcomes examined.

4. Power calculation: Did the authors perform a power calculation prior to the study? Is the study well-powered to detect differences in food choice based on genetic risk?

5. How representative is the sample? Supplementary Table 1 states that 80% of the participants are women, 70% have high physical activity and the average BMI is 27.9. What were the selection criteria for enrolling participants in the Choosewell study? Please discuss if there is potential selection bias in the study due to the participants not being selected at random but volunteering.

6. Quartiles: Why did the authors decide to summarise the polygenic scores into quartiles and present results as the comparison between highest and lowest quartile? How do the results look like when including the exact scores as quantitative predictors into the model?

Minor comments:

- Supplementary Figure 1. Please put "" around "ChooseWell 365", otherwise this may read as 365 study participants.

- Supplementary Table 2. Explain "highly expressed in the CNS" in the legend.

Reviewer #2: The authors present a secondary analysis(?) of an impressive trial, where participants health behaviour manipulated. The dataset is rich as objective food purchases have been measured, along with genetics. I would be very intrigued to read analysis of the phenotypic BMI in relation to the objective eating study variables. The PGS just seems to serve as a fancy method, but I think using PGS is an overkill - it makes associations very small and ultimately does not deliver any extra interpretative value. 

The study seems as a secondary analysis, as although the study is a trial, the manipulation is not being analysed. It would be helpful to have a reference to the paper analysing primary outcomes of the study, if available. The clinical trial outlines a moderating role of the PGS. This does not seem to be tested here. Please outline clearly, whether you followed the clinical trials protocol in this analysis or conducted exploratory additional analyses. Both approaches are ok, this just needs to be clear.

The study's goals fall short likely due to smallish sample size for these goals. This is expemplified by the use of quartiles throughout the paper, which highlight associations that likely do not emerge when using PGS-s as linear predictors. This approach has been previously critiqued in relation to polygenic scores. See this blog for accessible overview and further materials. https://discourse.datamethods.org/t/discussion-of-polygenic-prediction-of-weight-and-obesity-trajectories-from-birth-to-adulthood/1585 . If the authors wish to use quartiles, they should justify their choice in relation to this critique.

I invite authors to think through, what is the message and scientific question they want to deliver. If the goal is to understand how obesity influences food choices and reactiveness to the trial, then I would just use phenotypic obesity. This allows using a bigger sample, and also having bigger statistical effects. I would be very interested in learning the phenotypic results! 

Generally speaking, BMI_PGS-phenotype associations reproduce known BMI_phenotype-phenotype associations. So you could just study phenotypic associations and conclude that this is likely how genetic BMI plays out. Perhaps you could try reproducing the strongest phenotypic BMI- eating associations with a BMIpgs? 

If you want to make claims between BMIcns- and BMInon-cns, you need to test the statisical difference in effect sizes. http://comparingcorrelations.org/

Polygenic scores can also be used in certain conditions to infer causality. But I think that you need a larger sample size to do this kind of research. 

Doi.org/10.1002/sim.6835

Reviewer #3: This study addresses a more relevant aspect in the knowledge of determining factors of obesity that has also been scarcely described as is the choice of food.The work is well written and the results presented, below with some comments:

1.The authors mention the fact that is in workplace, it could be also interesting if they could discuss about what kind of food is available in those cafeteria and if that could be a bias

2.The authors mention that the collection was made three months retrospectively, is that because it was part of another study or why it is measured retrospectively?. In that sense the authors calculated the average?

3.It would be interesting if the authors could characterize the types of food available in the cafeteria, in addition to saying the traffic lights.

4.Although in complementary material they indicate the genes evaluated, it is interesting that in the discussion they could deepen into this and how some genes could effectively modulate aspects of food choice, put it in more concrete terms on the mechanism

[LINK]

---

## [Editor Report · Decision Letter 2]

4 Jun 2020

Dear Dr. Dashti,

Thank you very much for re-submitting your manuscript "Genetic risk for obesity and the quality, quantity, and timing of workplace food purchases: a cohort study" (PMEDICINE-D-20-00219R2) for review by PLOS Medicine.

I have discussed the paper with my colleagues and the academic editor and it was also seen again by reviewers. I am pleased to say that provided the remaining editorial and production issues are dealt with we are planning to accept the paper for publication in the journal.

[LINK]

We look forward to receiving the revised manuscript by Jun 11 2020 11:59PM. 

Sincerely,

Adya Misra, PhD

Senior Editor 

PLOS Medicine

plosmedicine.org

Requests from Editors:

Suggest title is tempered as you are not directly measuring genetic risk? In addition, the study descriptor might be better as ‘a secondary analysis from the ChooseWell 365 randomised trial’. 

Please provide a table with demographic information of participants

Please consider revising “influencing” to perhaps “associated with” as this is an observational study “We also assessed whether higher CNS-related obesity genetic risk that may be more likely to influence food behavioural choices than the non-CNS component”. The same goes for where you mention “causal in pathway”. I don’t think the study design or results allow this extrapolation, so please use more cautious language throughout. 

It appears that your STROBE checklist was not included in the revised manuscript and it was not called out in the methods section. Please do so, as requested previously. We ask that you don’t use line or page numbers as these are likely to change during publication

In the methods, in addition to simply mentioning that you had a prospective analysis plan, please provide this analysis plan as an SI file and also provide a call out to the file in the methods section. As stated previously “If a prospective analysis plan (from your funding proposal, IRB or other ethics

committee submission, study protocol, or other planning document written

before analyzing the data) was used in designing the study, please include the

relevant prospectively written document with your revised manuscript as a

Supporting Information file to be published alongside your study, and cite it in

the Methods section. A legend for this file should be included at the end of your manuscript”. 

Please add unaccounted confounding as a limitation in your discussion, as there are several confounders linked to food choice and not all can be measured in an observational setting.

Comments from Reviewers:

[LINK]

---

## [Editor Report · Decision Letter 3]

18 Jun 2020

Dear Dr. Dashti, 

On behalf of my colleagues and the academic editor, Dr. Sanjay Basu, I am delighted to inform you that your manuscript entitled "Polygenic risk score for obesity and the quality, quantity, and timing of workplace food purchases: a secondary analysis from the ChooseWell 365 randomised trial" (PMEDICINE-D-20-00219R3) has been accepted for publication in PLOS Medicine. 

PRODUCTION PROCESS

PRESS

PROFILE INFORMATION

Thank you again for submitting the manuscript to PLOS Medicine. We look forward to publishing it. 

Best wishes, 

Adya Misra, PhD

Senior Editor 

PLOS Medicine

plosmedicine.org